

# Circulating HMGB1 in acute ischemic stroke and its association with post-stroke cognitive impairment

Zhenbao Liu[1,*], Weixia Yang[1,*], Jianxin Chen[2] and Qian Wang[3]

[1] Department of Neurology, Qingpu Branch of Zhongshan Hospital, Fudan University, Shanghai, China
[2] Department of Critical Care Medicine, Jinan First People's Hospital, Shandong Traditional Chinese Medicine University, Jinan, Shandong, China
[3] Department of Central Laboratory, The Affiliated Taian City Central Hospital of Qingdao University, Taian, Shandong, China
[*] These authors contributed equally to this work.

Corresponding authors
Jianxin Chen, cjxneurology@sina.com
Qian Wang, qianqian-wangxi@163.com

## ABSTRACT

**Background**. Ischemic stroke frequently leads to a condition known as post-stroke cognitive impairment (PSCI). Timely recognition of individuals susceptible to developing PSCI could facilitate the implementation of personalized strategies to mitigate cognitive deterioration. High mobility group box 1 (HMGB1) is a protein released by ischemic neurons and implicated in inflammation after stroke. Circulating levels of HMGB1 could potentially serve as a prognostic indicator for the onset of cognitive impairment following ischemic stroke.

**Objective**. To investigate the predictive value of circulating HMGB1 concentrations in the acute phase of ischemic stroke for the development of cognitive dysfunction at the 3-month follow-up.

**Methods**. A total of 192 individuals experiencing their initial episode of acute cerebral infarction were prospectively recruited for this longitudinal investigation. Concentrations of circulating HMGB1 were quantified using an enzyme-linked immunosorbent assay (ELISA) technique within the first 24 hours following hospital admission. Patients underwent neurological evaluation including NIHSS scoring. Neuropsychological evaluation was conducted at the 3-month follow-up after the cerebrovascular event, employing the Montreal Cognitive Assessment (MoCA) as the primary tool for assessing cognitive performance. Multivariable logistic regression models were employed to investigate the relationship between circulating HMGB1 concentrations and cognitive dysfunction following stroke, which was operationalized as a MoCA score below 26, while controlling for potential confounders including demographic characteristics, stroke severity, vascular risk factors, and laboratory parameters.

**Results**. Of 192 patients, 84 (44%) developed PSCI. Circulating HMGB1 concentrations were significantly elevated in individuals who developed cognitive dysfunction following stroke compared to those who maintained cognitive integrity ($8.4 \pm 1.2$ ng/mL vs $4.6 \pm 0.5$ ng/mL, respectively; $p < 0.001$). The prevalence of PSCI showed a dose-dependent increase with higher HMGB1 quartiles. After controlling for potential confounders such as demographic factors (age, gender, and education), stroke severity, vascular risk factors, and laboratory parameters in a multivariable logistic regression model, circulating HMGB1 concentrations emerged as a significant independent predictor of cognitive dysfunction following stroke (regression coefficient = 0.236, $p < 0.001$).

**Conclusion**. Circulating HMGB1 concentrations quantified within the first 24 hours following acute cerebral infarction are significantly and independently correlated with the likelihood of developing cognitive dysfunction at the 3-month follow-up, even after accounting for potential confounding factors. HMGB1 may be a novel biomarker to identify patients likely to develop post-stroke cognitive impairment for targeted preventive interventions.

## INTRODUCTION

Post-stroke cognitive impairment (PSCI) is increasingly recognized as a major health issue, affecting up to 64% of stroke survivors (*Cumming & Brodtmann, 2011*; *Chen et al., 2022*). However, our ability to predict cognitive outcomes after stroke remains limited (*Lees et al., 2014*). Identifying blood-based biomarkers associated with poststroke cognitive decline is critically needed to allow early identification of patients at risk, guide treatment decisions, and shed light on underlying mechanisms (*Kisler et al., 2017*). Serum biomarkers offer key advantages as they can be measured easily in routine clinical practice using standardized assays (*Miao & Liao, 2014*). Increased blood–brain barrier permeability after ischemia allows brain-derived proteins like neurofilament light and tau to enter the bloodstream (*Skillbäck et al., 2014*). Elevated serum levels of these neuronal damage markers after stroke may predict more severe damage and a higher risk of later cognitive problems (*Hesse et al., 2000*). Establishing reliable serum biomarkers for poststroke cognitive impairment will accelerate prognostication, enable early targeted interventions, and ultimately improve functional outcomes in this vulnerable patient population.

High mobility group box 1 (HMGB1) has been identified as a crucial player in orchestrating detrimental inflammatory responses in various acute cerebral insults, including stroke, and has also been implicated in the pathogenesis of several neurodegenerative disorders. HMGB1 is a ubiquitous protein found in the nuclear and cytoplasmic compartments of virtually all cell types, playing a crucial role in the regulation of gene expression and the maintenance of chromatin architecture under homeostatic conditions (*Chen et al., 2019*; *Sun et al., 2019*). However, during states of cellular damage or stress, HMGB1 can be actively secreted or passively released into the extracellular space where it triggers inflammatory pathways. Ischemic stroke leads to extensive HMGB1 release from dying neurons in the infarct core starting within 30 min of onset, followed by a delayed second wave of active HMGB1 secretion from reactive astrocytes and microglia that can persist for weeks (*Hayakawa, Qiu & Lo, 2010*; *Chen et al., 2019*). In both experimental animal stroke models and clinical studies involving human subjects, elevated peripheral blood concentrations of HMGB1 have been observed following acute cerebral ischemia, which is thought to reflect the ongoing neuroinflammatory processes triggered by the ischemic insult (*Liu et al., 2007*; *Muhammad et al., 2008*). There is growing evidence from preclinical rodent studies that extracellular HMGB1 drives maladaptive pro-inflammatory

processes during the acute phase of ischemia through receptors like RAGE, TLR2, and TLR4 which lead to expansion of infarct size and worsened functional outcomes (*Muhammad et al., 2008*; *Hayakawa, Qiu & Lo, 2010*). HMGB1 is also implicated as a key mediator of chronic neuroinflammation, white matter injury, neurovascular unit disruption, and network dysfunction in the weeks to months after an initial stroke which may contribute to secondary neurodegeneration and associated cognitive impairment (*Sumbria, Boado & Pardridge, 2012*). While some studies have suggested an association between higher circulating concentrations of HMGB1 and a greater likelihood of cognitive deterioration following stroke in human subjects, the evidence remains inconclusive due to conflicting results reported in the literature (*Yang et al., 2010*). Elucidating the specific mechanisms by which HMGB1 signaling triggers subacute and long-term neuroinflammation that may lead to post-stroke cognitive deficits is an important avenue for the development of targeted therapies to improve functional outcomes after ischemic brain injury.

Despite emerging evidence linking early elevated HMGB1 levels with the risk of post-stroke cognitive dysfunction, most prior studies had small sample sizes and measured HMGB1 beyond the initial 24-hour period when neuroinflammatory processes rapidly escalated. Additionally, detailed cognitive assessments were often lacking. Whether very early HMGB1 levels are associated with later development of post-stroke cognitive impairment requires further investigation. This study aimed to address critical gaps in knowledge by evaluating serum HMGB1 levels drawn within 24 h of ischemic stroke as a predictor of cognitive impairment determined by validated neuropsychological screening at 3-month follow-up. We hypothesized that acute elevations in systemic HMGB1 would independently predict the risk of post-stroke cognitive decline even after accounting for potential confounding factors. Elucidating the prognostic utility of ultra-early HMGB1 for the development of cognitive sequelae after ischemic stroke is an important step toward identifying at-risk patients for targeted monitoring and preventative therapies, as well as unraveling the complex mechanisms of post-stroke neuroinflammation and secondary neurodegeneration.

## MATERIAL AND METHODS

### Study participants

This longitudinal investigation recruited individuals hospitalized at the Qingpu Branch of Zhongshan Hospital due to an initial episode of acute cerebral infarction, which was corroborated by neuroimaging findings. Inclusion criteria were age ≥18 years, stroke onset <7 days before admission, and no prior dementia/cognitive impairment. Patients were excluded if they had hemorrhagic stroke, severe aphasia, inability to complete cognitive testing, history of central nervous system disease, presence of infection/inflammatory or autoimmune disorder, chronic liver/kidney disease, alcoholism, substance abuse, use of medications affecting cognition, lacked serum HMGB1 measurement within 24 h, or could not complete 3-month follow-up. Prior to enrollment, all subjects or their legally designated surrogates provided written consent after being fully informed about the study. The investigation was conducted in compliance with the ethical principles outlined in the

Declaration of Helsinki and received approval from the Institutional Review Board of Jinan First People's Hospital (reference number: JNFH-20210069).

Figure 1 employed a schematic diagram to illustrate the subject recruitment and inclusion criteria for a longitudinal investigation assessing the predictive value of circulating HMGB1 concentrations in the development of cognitive dysfunction following acute ischemic stroke. The flowchart starts with 315 patients with first-ever acute ischemic stroke screened for eligibility. 18 patients were excluded for reasons like hemorrhagic stroke, severe aphasia, inability to undergo cognitive testing, prior central nervous system disease, active infections, chronic organ disease, and substance abuse. Another 105 patients were excluded due to lack of blood sample for HMGB1 testing, loss to follow-up, withdrawal, new neurological/psychiatric illnesses, or use of medications affecting cognition. The remaining 192 eligible patients were enrolled and underwent baseline assessments including NIHSS scores, brain imaging, laboratory tests, and measurement of serum HMGB1 levels over 24 h. Neuropsychological evaluation was conducted at the 3-month post-stroke time point, employing the Montreal Cognitive Assessment (MoCA) as the primary tool for measuring cognitive performance. MOCA score <26 was categorized as post-stroke cognitive impairment (PSCI) group ($n = 84$), and MOCA score $\geq 26$ was categorized as non-PSCI group ($n = 108$). Lastly, circulating HMGB1 concentrations were contrasted between individuals who developed cognitive dysfunction following stroke and those who maintained cognitive integrity to evaluate the potential of this biomarker in forecasting the occurrence of neuropsychological deficits in the post-stroke period.

## Baseline characteristics

Upon inclusion in the study, demographic and background information was gathered for each participant. The data collected encompassed factors such as chronological age, biological sex, and the extent of formal schooling. The demographic and clinical data of each enrolled subject were meticulously documented by trained personnel to facilitate subsequent statistical evaluation. Baseline data collection included demographics, stroke severity (NIHSS score), vascular risk factors, lab results, and serum HMGB1 levels within 24 h of admission. Follow-up at 3 months assessed cognitive function using the Montreal Cognitive Assessment.

## Blood samples test

Within the first 24 h of hospitalization, a sample of fasting peripheral venous blood was obtained from each participant. The blood collected in the serum separation tube (SST) was allowed to clot at room temperature for 30 min before being centrifuged at $1,000 \times g$ for a quarter of an hour. All participants were instructed to fast for at least 12 h prior to blood sample collection, abstaining from food and beverages other than water during this period. The serum from each subject was either used immediately for analysis or divided into aliquots and stored at $-20$ °C for future use. The levels of HMGB1 in the serum were quantified using an enzyme-linked immunosorbent assay (ELISA) kit (Abcam, Cambridge, MA, USA) following the protocol provided by the manufacturer. In summary, the assay involved the following steps: first, microtiter plates pre-coated with capture antibodies

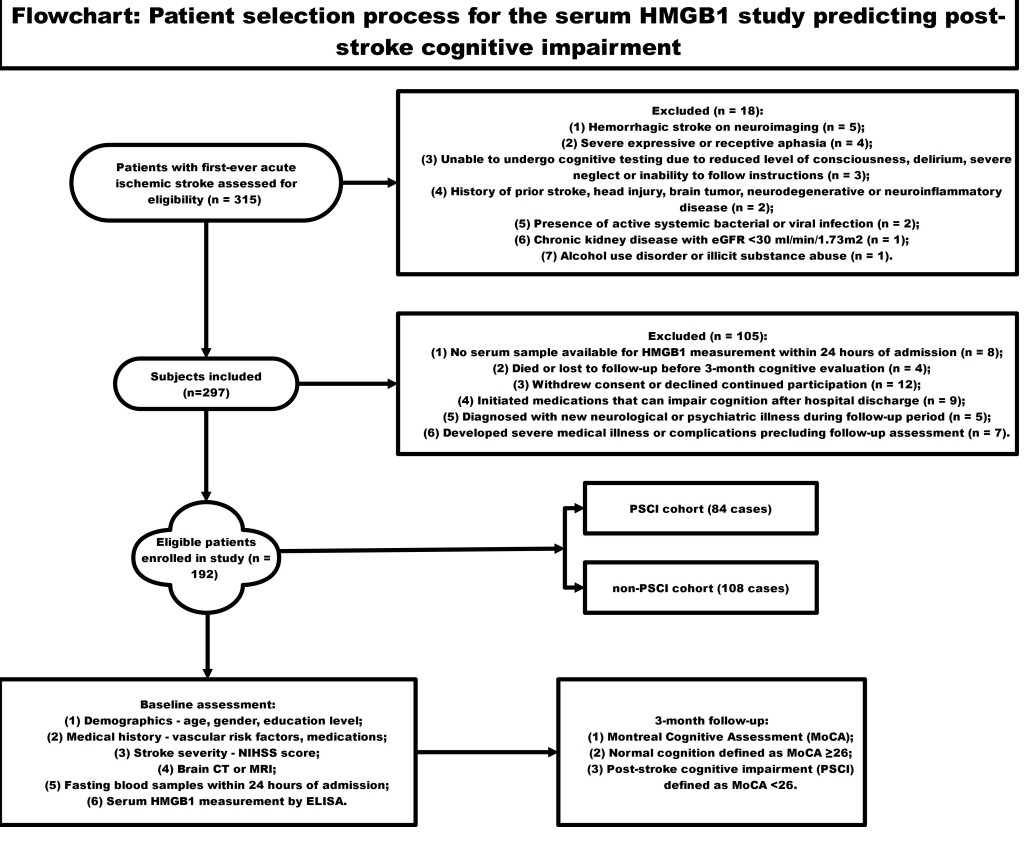

**Figure 1** The flowchart showing the patient selection process for the serum HMGB1 study predicting post-stroke cognitive impairment.

were incubated with standards and appropriately diluted serum samples in assay buffer for 2 h at ambient temperature. After a washing step, the plates were then incubated with HRP-conjugated detection antibodies for 2 h, followed by a 1-hour incubation with streptavidin-HRP. TMB substrate was added for color development which was stopped using 2N H2SO4. Absorbance was measured at 450nm. Standard curves were plotted using 4-parameter logistic regression and sample concentrations were interpolated from the standard curve. Before the addition of samples, capture antibodies were diluted in carbonate-bicarbonate buffer and incubated at 4 °C for 12 h. Subsequently, the plates were blocked with PBS containing 1% BSA for 60 min. Following each incubation step, the plates were washed thrice using PBS-Tween 20 wash buffer with the aid of an automated plate washing system. All samples were assayed in duplicate and average values were reported.

Other clinical laboratory tests were assayed from blood samples using standard methods. For example, fasting blood glucose (FBG) was measured using the hexokinase method on an automated analyzer. The concentrations of high-density lipoprotein (HDL) and low-density lipoprotein (LDL) cholesterol, as part of the lipid panel, were quantified employing enzyme-based colorimetric techniques performed on an automated analytical system. Complete blood count including hemoglobin and leukocyte count was analyzed using

a hematology analyzer. Following a 5-minute period of seated rest, both systolic blood pressure (SBP) and diastolic blood pressure (DBP) were measured using a conventional mercury-based sphygmomanometer. All clinical laboratory tests were performed by experienced technicians with no knowledge of the clinical data. Commercial quality control materials were used to validate test procedures and reliability.

## Neurological assessment

The severity of the stroke was evaluated by a minimum of two qualified neurologists within the first 24 h of hospitalization, employing the National Institutes of Health Stroke Scale (NIHSS) as the primary assessment tool. The NIHSS is a comprehensive assessment tool that assesses the degree of neurological deficit across 11 distinct categories, which include level of consciousness, gaze, visual fields, facial paralysis, motor function, coordination, sensory perception, language, speech articulation, and spatial attention. Each domain scores between 0–4 with 0 being normal and higher scores indicating greater deficit severity. Total NIHSS scores range from 0–42 with scores of 0–6 indicating mild stroke, 7–15 moderate stroke, and 15–42 severe stroke. Certified raters followed standardized instructions for administering the NIHSS. Scores were based on observed performance and neurological examination. The NIHSS took approximately 5–10 min to administer and score. To minimize potential biases associated with the subjective nature of the NIHSS, all neurological assessments were performed by trained and certified neurologists. The neurologists underwent standardized training to ensure consistency in the administration and scoring of the NIHSS. Regular quality control measures, such as inter-rater reliability assessments, were conducted to maintain high levels of consistency across evaluators.

## Cognitive assessment

Neuropsychological status was assessed at the 3-month follow-up after the cerebrovascular event by a minimum of two qualified neuropsychologists, employing the Montreal Cognitive Assessment (MoCA) as the primary evaluation tool. The MoCA is a comprehensive cognitive screening tool that evaluates performance across eight distinct neuropsychological domains, utilizing a total of 30 items. These domains and their corresponding maximum scores include: visuospatial/executive abilities (5 points), object naming (3 points), memory encoding (5 points), attentional capacity (6 points), language processing (3 points), abstract reasoning (2 points), delayed memory retrieval (5 points), and spatial and temporal orientation (6 points). Each item scores 1 point for correct responses with a total possible score of 30. In accordance with established criteria, individuals who obtained scores of 26 or higher on the MoCA were classified as having intact cognitive function, whereas those with scores below the 26-point threshold were considered to have cognitive deficits, indicative of post-stroke cognitive impairment (PSCI). An extra point was given to subjects with $\leq 12$ years of education to account for educational bias. The MoCA examination took approximately 10–15 min to administer. Standardized instructions were provided in the subjects' native languages using validated translated MoCA versions. Subjects were allowed to use necessary assistive devices like hearing aids and glasses. We have obtained the written permission to use the MoCA from

the copyright holder. To reduce potential biases related to the subjective nature of the MoCA, all cognitive assessments were conducted by experienced neuropsychologists who were blinded to the participants' clinical information. The neuropsychologists received standardized training in the administration and scoring of the MoCA to ensure consistency. Regularly scheduled calibration sessions were held to maintain inter-rater reliability and minimize variability in scoring.

## Logistic regression analysis

To assess the association between serum HMGB1 levels and risk of post-stroke cognitive impairment, multivariate logistic regression models were constructed with PSCI as the dependent variable. First, a foundational model was constructed, accounting for demographic characteristics (age, sex, educational attainment), cardiovascular risk factors (elevated blood pressure, diabetes mellitus, tobacco use), and the extent of neurological deficits as assessed by the NIHSS. A second model with additional adjustment for lipid levels was then created to account for potential confounding by these metabolic factors. Finally, a fully adjusted model controlling for all potential confounders was created. This included the demographic factors, vascular risk factors, NIHSS score, lipid levels as well as other laboratory parameters like hemoglobin and leukocyte counts. Serum HMGB1 was modeled both as a continuous variable and categorically by quartiles in the regression analyses to assess for dose-dependent relationships. To quantify the relationship between circulating HMGB1 concentrations and the likelihood of developing post-stroke cognitive impairment, odds ratios and their corresponding 95% confidence intervals were derived. Statistical significance was set at a threshold of 0.05 for two-tailed tests. The logistic regression models were constructed and analyzed using the 23th iteration of the Statistical Package for the Social Sciences (SPSS) software (IBM Corp., Armonk, NY, USA).

## Statistical analysis

Continuous data were presented as mean $\pm$ standard deviation for variables following a normal distribution, while those with a non-normal distribution were reported as median (interquartile range). Categorical data were expressed as percentages. Comparisons between groups were performed using the following statistical tests: Student's $t$-test for normally distributed continuous variables, Mann–Whitney U test for continuous variables with a skewed distribution, and chi-square test for categorical variables. Statistical analyses were conducted using the 23rd version of the Statistical Package for the Social Sciences (SPSS) software (IBM Corp., Armonk, NY, USA). The threshold for statistical significance was set at a $p$-value below 0.05 for two-sided tests.

# RESULTS

## Baseline assessment of patients' cognitive functional capacities

The present cohort study encompassed 192 individuals diagnosed with acute ischemic stroke (AIS). At baseline, we documented the following patient attributes: chronological age, biological sex, years of formal schooling, systolic blood pressure, diastolic blood pressure, fasting blood glucose levels, high-density lipoprotein cholesterol, low-density lipoprotein

**Table 1  Baseline information based on patients' cognitive function.**

|  | PSCI ($n = 84$) | Non-PSCI ($n = 108$) | P |
|---|---|---|---|
| Age, years | $67.1 \pm 6.0$ | $66.8 \pm 7.4$ | 0.763 |
| Gender, male, n (%) | 56 (66.7) | 77 (71.3) | 0.490 |
| Formal education, years | $9.2 \pm 1.4$ | $9.4 \pm 1.5$ | 0.347 |
| SBP, mmHg | $136.8 \pm 5.9$ | $136.3 \pm 6.1$ | 0.568 |
| DBP, mmHg | $89.7 \pm 6.5$ | $89.2 \pm 6.8$ | 0.607 |
| FBG, mmol/L | $6.3 \pm 0.8$ | $6.1 \pm 0.9$ | 0.111 |
| HDL, mmol/L | $1.3 \pm 0.2$ | $1.2 \pm 0.3$ | 0.009 |
| LDL, mmol/L | $2.6 \pm 0.4$ | $2.5 \pm 0.5$ | 0.003 |
| Hemoglobin, g/L | $128.3 \pm 11.2$ | $127.9 \pm 11.7$ | 0.811 |
| Leukocyte, $10^9$/L | $6.8 \pm 0.6$ | $6.7 \pm 0.8$ | 0.341 |
| MoCA, points | $23.2 \pm 1.7$ | $27.4 \pm 1.3$ | <0.001 |
| HMGB1, ng/mL | $8.4 \pm 1.2$ | $4.6 \pm 0.5$ | <0.001 |

**Notes.**

PSCI, poststroke cognitive impairment; SBP, systolic blood pressure; DBP, diastolic blood pressure; FBG, fasting blood glucose; HDL, high-density lipoprotein; LDL, low-density lipoprotein; MoCA, Montreal Cognitive Assessment; HMBG1, High Mobility Group Protein B1.

cholesterol, hemoglobin concentration, leukocyte count, Montreal Cognitive Assessment (MoCA) scores, and high mobility group box 1 (HMGB1) levels. At the three-month mark following the acute ischemic stroke (AIS) event, we utilized the Montreal Cognitive Assessment (MoCA) instrument to assess cognitive functioning across the entire AIS patient cohort. Based on their MoCA performance scores, we stratified the AIS patients into two distinct subgroups: those exhibiting post-stroke cognitive impairment (PSCI cohort, $n = 84$) and those without cognitive deficits (non-PSCI cohort, $n = 108$). Upon contrasting the baseline demographic and clinical profiles between the two subgroups, our analysis revealed no discernible differences that attained statistical significance ($p > 0.05$) across the array of parameters evaluated. The montreal cognitive assessment (MoCA) score in the PSCI group ($23.2 \pm 1.7$) was significantly lower compared to that in the non-PSCI group ($27.4 \pm 1.3$). Quantification of circulating HMGB1 concentrations unveiled markedly elevated serum levels of this inflammatory mediator in the PSCI subgroup, registering at ($8.4 \pm 1.2$) ng/mL, as opposed to the non-PSCI counterparts who exhibited comparatively lower values of ($4.6 \pm 0.5$) ng/mL. Our analysis unveiled a remarkably higher circulating HMGB1 burden among individuals in the PSCI subgroup when contrasted against their non-cognitively impaired counterparts, with this disparity in serum levels proving to be statistically significant ($p < 0.001$). The demographic and clinical characteristics of the acute ischemic stroke (AIS) patient cohort at study entry are comprehensively delineated in Table 1.

## Association between circulating HMGB1 burden and post-stroke cognitive outcomes

Table 2 depicts the incidence rates of PSCI stratified according to quartiles of serum high mobility group box 1 (HMGB1) concentrations observed within our cohort. Our analyses unveiled a distinct linear trend, wherein the risk of developing post-stroke cognitive

**Table 2  Correlation between prevalence of PSCI and serum HMBG1 levels.**

| Variable | PSCI, n (%) | | | | p for trend |
|---|---|---|---|---|---|
| | Q1 ($n = 48$) | Q2 ($n = 48$) | Q3 ($n = 48$) | Q4 ($n = 48$) | |
| PSCI, n (%) | 13 | 17 | 23 | 31 | <0.05 |

Notes.
PSCI, poststroke cognitive impairment; HMBG1, High Mobility Group Protein B1.

**Table 3  Regression analysis of correlation between serum HMGB1 levels and PSCI.**

| Variables | Regression coefficient | *P* values |
|---|---|---|
| Model 1 | 0.314 | <0.001 |
| Model 2 | 0.287 | <0.001 |
| Model 3 | 0.236 | <0.001 |

Notes.
Model 1 adjusted for age, gender, formal education, SBP, DBP, FBG, Hemoglobin and Leukocyte. Model 2 adjusted for model 1, HDL and LDL. Model 3 adjusted for model 2 and HMGB1.
HMBG1, High Mobility Group Protein B1; PSCI, poststroke cognitive impairment; SBP, systolic blood pressure; DBP, diastolic blood pressure; FBG, fasting blood glucose; HDL, high-density lipoprotein; LDL, low-density lipoprotein.

impairment escalated in lockstep with higher baseline circulating HMGB1 levels across the defined quartile strata among acute ischemic stroke patients ($p < 0.001$).

## Logistic regression modeling to evaluate the link between HMGB1 and post-stroke cognitive impairment

To investigate the potential utility of circulating HMGB1 as a prognostic biomarker for post-stroke cognitive impairment (PSCI), we conducted multivariate logistic regression modeling adjusted for relevant covariates. The results emanating from the logistic regression analyses are comprehensively delineated in Table 3. Even after accounting for potential confounding influences of age, gender, years of formal education, systolic and diastolic blood pressure, fasting blood glucose, hemoglobin, and leukocyte counts in Model 1, elevated serum HMGB1 levels emerged as an independent risk predictor for post-stroke cognitive impairment (regression coefficient = 0.314, $p < 0.001$). Additionally, in Model 2, which incorporated HDL and LDL cholesterol levels as covariates in addition to the variables adjusted for in Model 1, elevated circulating HMGB1 continued to exhibit a robust independent association with an increased risk of post-stroke cognitive impairment (regression coefficient = 0.287, $p < 0.001$). Even in the fully adjusted Model 3, which accounted for the covariates incorporated in Model 2 as well as HMGB1 levels, elevated circulating HMGB1 retained its status as an independent risk predictor for post-stroke cognitive impairment (regression coefficient = 0.236, $p < 0.001$). Collectively, these findings underscore the robust association between heightened HMGB1 burden and an increased vulnerability to developing cognitive deficits following acute ischemic stroke.

## DISCUSSION

In this prospective cohort investigation, we discovered that serum concentrations of HMGB1 assessed within the first 24 h following acute ischemic stroke onset were correlated with an increased likelihood of developing cognitive dysfunction at the 3-month follow-up.

Among 192 patients with first-ever ischemic stroke, elevated HMGB1 levels early after stroke were an independent predictor of poorer cognitive performance on the Montreal Cognitive Assessment at 3-month follow-up. The study revealed a direct proportional association between HMGB1 concentrations and the incidence of cognitive dysfunction, with elevated levels correlating to a heightened probability of experiencing cognitive deficits. The results of this study indicate that HMGB1 could potentially serve as an innovative biological marker to pinpoint individuals who are more susceptible to experiencing cognitive deterioration following a stroke, and who might gain from timely therapeutic interventions.

The key discovery that heightened HMGB1 concentrations in the acute phase following ischemic stroke correlate with an amplified probability of developing PSCI is consistent with and expands upon the findings reported by multiple prior investigations. Several previous studies have described associations between circulating HMGB1 levels measured during the initial days to weeks following a cerebrovascular accident and the subsequent emergence of cognitive impairments (*Liu et al., 2007*; *Yang et al., 2010*; *Tian et al., 2017*; *Qiu et al., 2023*). For example, *Shan et al. (2022)* discovered that HMGB1 concentrations assessed at the 72-hour mark post-stroke were associated with performance on the Mini-Mental State Exam (MMSE) administered 3 months later in a cohort of 56 individuals who had experienced a cerebrovascular event. Furthermore, heightened levels of HMGB1 have been implicated in the development of cognitive deficits in preclinical studies utilizing animal models of ischemic brain injury (*Muhammad et al., 2008*), while HMGB1 inhibition resulted in improved memory and learning post-stroke in rodents (*Mazarati et al., 2011*). Moreover, HMGB1 has been implicated in the pathogenesis of cognitive impairments across a spectrum of neurological conditions, including epileptic disorders, Alzheimer's disease, and various other neurodegenerative processes (*Yang et al., 2015*; *Paudel et al., 2019*). From a mechanistic standpoint, HMGB1 is recognized as a potent activator of neuroinflammatory cascades that compromise the blood–brain barrier's structural and functional integrity, ultimately resulting in white matter injury (*Hayakawa, Qiu & Lo, 2010*; *Sumbria, Boado & Pardridge, 2012*). HMGB1 levels positively correlated with markers of neural damage like S100B and NSE in stroke patients (*Oda et al., 2012*). The temporal relationship and dose–response effects observed in this study provide added evidence that HMGB1 may play a causal role in mechanisms underlying PSCI.

HMGB1 has been strongly implicated in triggering neuroinflammatory responses that lead to neuronal injury and cognitive decline after ischemic stroke through multiple signaling mechanisms. HMGB1 engages with receptors such as RAGE, TLR2, and TLR4, thereby triggering the activation of microglia and astrocytes and inducing the secretion of proinflammatory mediators, including TNF-$\alpha$, IL-1$\beta$, IL-6, and chemotactic factors like CCL2, CXCL8, and CXCL12 (*Muhammad et al., 2008*; *Hayakawa, Qiu & Lo, 2010*). Downstream activation of NF-kB, MAPK and JAK/STAT pathways further amplifies this inflammatory response (*Muhammad et al., 2008*; *Xie et al., 2019*). HMGB1 also increases matrix metalloproteinases like MMP-2 and MMP-9 which disrupt blood–brain barrier integrity, contributing to vasogenic edema and worsened ischemia (*Hayakawa, Qiu & Lo, 2010*; *Li et al., 2013*). At the level of the neurovascular interface, HMGB1 modulates the expression profile of critical tight junction components, such as occludin, claudin-5, and

ZO-1, while simultaneously upregulating aquaporin-4 channel expression, culminating in a disruption of the blood–brain barrier's selective permeability (*Qiu et al., 2008*; *Li et al., 2013*). In the chronic phase following a cerebrovascular accident, HMGB1 exerts detrimental effects on synaptic plasticity by attenuating long-term potentiation through its influence on NMDA receptor function, dendritic spine morphology, and neural networks that subserve learning and memory processes (*Costello et al., 2011*; *Paudel et al., 2018*). HMGB1 inhibits hippocampal neurogenesis through RAGE/TLR4 signaling which limits cognitive recovery (*Lei et al., 2013*). Sustained neuroinflammation mediated by HMGB1 also causes white matter injury by inducing astrocyte reactivity, microglial activation, and oligodendrocyte death (*Sumbria, Boado & Pardridge, 2012*). Therefore, HMGB1 triggers both acute cytotoxic effects as well as chronic network dysfunction through neuroplasticity and neurogenesis changes that may result in PSCI.

While our study focused on the relationship between HMGB1 and PSCI, it is important to consider the potential interactions of HMGB1 with broader neuroinflammatory processes in the context of stroke and cognitive impairment. HMGB1 is known to be a key mediator of inflammation, and its release from damaged neurons and activated immune cells can trigger a cascade of inflammatory responses in the brain (*Singh et al., 2016*; *Yang, Andersson & Brines, 2021*). In the setting of ischemic stroke, HMGB1 has been shown to interact with various pattern recognition receptors, such as Toll-like receptors (TLRs) and the receptor for advanced glycation end products (RAGE), leading to the activation of inflammatory signaling pathways (*Ye et al., 2019*). The activation of these pathways results in the production of pro-inflammatory cytokines, chemokines, and other mediators that can exacerbate neuronal damage and contribute to the development of cognitive impairment (*Squillace & Salvemini, 2022*; *Naz et al., 2023*). Moreover, HMGB1 has been implicated in the disruption of the blood–brain barrier and the recruitment of peripheral immune cells to the brain, further amplifying the neuroinflammatory response (*Paudel et al., 2020*; *Nishibori et al., 2020*). These processes can lead to persistent neuroinflammation, which has been increasingly recognized as a key driver of cognitive decline in various neurological conditions, including stroke (*Cheng et al., 2022*). Future studies should aim to elucidate the complex interactions between HMGB1 and other inflammatory mediators in the pathogenesis of PSCI. Understanding these relationships may provide valuable insights into potential therapeutic targets for mitigating neuroinflammation and improving cognitive outcomes after stroke.

HMGB1 has emerged as a critical player in orchestrating inflammatory responses and propagating tissue damage across a diverse array of neurological conditions that extend beyond cerebrovascular events, encompassing traumatic brain injury, seizure disorders, Alzheimer's disease, Parkinson's disease, motor neuron disease, and demyelinating disorders such as multiple sclerosis (*Yang et al., 2005*; *Juranek et al., 2013*; *Lee et al., 2014*; *Chen et al., 2019*; *Dai et al., 2021*). In many of these conditions, HMGB1 acts as an early pro-inflammatory signal released from damaged neurons that triggers and sustains chronic neuroinflammatory responses (*Yang et al., 2005*; *Dai et al., 2021*). In preclinical studies utilizing animal models of acute seizures and chronic epilepsy, HMGB1 has been implicated in both the initiation of individual seizure events (ictogenesis) and the progressive

alterations that lead to the emergence of spontaneous and recurrent epileptic activity (epileptogenesis), exerting its influence through modulation of neuronal excitability and synaptic plasticity mechanisms (*Maroso et al., 2010*; *Dai et al., 2021*). In neurodegenerative diseases like Alzheimer's and Parkinson's, HMGB1 release from dying neurons may activate microglia and propagate sustained inflammation that drives progressive neurodegeneration (*Lee et al., 2014*). Preclinical studies investigating a range of neurological conditions have highlighted the potential of therapeutic interventions aimed at modulating HMGB1 release or disrupting its interaction with receptors such as RAGE, demonstrating promising results in attenuating neuronal injury and ameliorating cognitive and behavioral impairments (*Muhammad et al., 2008*; *Kim et al., 2011*; *Dai et al., 2021*). Therefore, HMGB1 represents an important convergent mechanism of neural injury across diverse acute and chronic neurologic conditions that may be a fruitful target for novel neuroprotective treatments.

PSCI is increasingly recognized to have a relationship with neuroinflammation. PSCI is a condition characterized by cognitive impairments that can occur after surgery, affecting memory, attention, and executive functions. The relationship between PSCI and neuroinflammation highlights the impact of systemic inflammation on brain function and cognition, suggesting that strategies to mitigate neuroinflammation could be beneficial in preventing or treating PSCI. This investigation contributes significant advancements to the field by being among the pioneering studies to specifically assess the utility of HMGB1 as a predictive biomarker for cognitive dysfunction following acute ischemic stroke. While HMGB1 has been extensively studied for its role in mediating acute ischemic injury (*Liu et al., 2007*; *Hayakawa, Qiu & Lo, 2010*), its utility in predicting longer-term cognitive outcomes after stroke has been limited thus far. Only a handful of previous studies have examined associations between peripheral HMGB1 levels and cognitive dysfunction in stroke patients, with inconsistent results (*Yang et al., 2010*; *Shan et al., 2022*). Additionally, the majority of prior studies measured HMGB1 beyond the initial 24-hour window (*Sun et al., 2014*), whereas this study captured ultra-acute HMGB1 levels. Demonstrating that very early HMGB1 predicts later cognitive trajectory is a key novelty. A further novel aspect of this study is the employment of the Montreal Cognitive Assessment, a well-established and standardized cognitive screening instrument, to systematically evaluate post-stroke cognitive function. The prospective cohort design, large sample size, and rigorous control for confounders also strengthen evidence over previous works. This study helps address major gaps in prognostic biomarkers to identify stroke patients at risk of dementia and provides a rationale for targeting HMGB1 therapeutically to prevent post-stroke cognitive decline. Additional research is warranted to unravel the precise causal pathways through which HMGB1 contributes to the development of neurocognitive consequences following cerebrovascular accidents. Overall, this study represents an important advance in guiding the prognosis and management of the increasingly prevalent and disabling consequences of stroke.

While this study provides novel evidence for HMGB1 as a prognostic biomarker for PSCI, there are some limitations to consider regarding the study design. As this was a prospective cohort study, it cannot prove causality between elevated HMGB1 and the development of PSCI. Although we adjusted for multiple potential confounders in our

statistical analyses, the influence of unmeasured or residual confounding factors cannot be completely ruled out. Furthermore, the cognitive assessment was only performed at a single time point, which limits the interpretation of the temporal relationship between HMGB1 and cognitive decline. These inherent limitations of observational studies should be taken into account when interpreting the results and inferring causality. Future research should focus on conducting randomized controlled trials to investigate the effect of interventions targeting HMGB1 on cognitive outcomes after stroke. These inherent limitations of observational studies should be taken into account when interpreting the results and inferring causality. Future research should focus on conducting randomized controlled trials to investigate the effect of interventions targeting HMGB1 on cognitive outcomes after stroke. Additionally, mechanistic studies are needed to elucidate the underlying biological pathways linking HMGB1 to the development of cognitive impairment following stroke.

It is important to acknowledge the limitations of our single-center design and relatively small sample size. Furthermore, the lack of a validation cohort in our study limits the immediate generalizability and clinical applicability of our findings. Future validation studies should ideally employ similar methodologies, including standardized cognitive assessments and timepoints, to ensure comparability with our study. To further validate our findings and establish the robustness of HMGB1 as a predictor of cognitive outcomes after stroke, it is crucial to conduct larger, multi-center studies with diverse patient populations. Such studies will help to assess the generalizability of our results and provide a more comprehensive understanding of the relationship between HMGB1 and post-stroke cognitive impairment. Moreover, replication studies in independent cohorts are necessary to confirm the reliability and reproducibility of our findings. By validating our results in different settings and populations, we can strengthen the evidence supporting the use of HMGB1 as a prognostic tool in clinical practice. Future studies should also consider exploring the predictive value of HMGB1 in combination with other biomarkers and clinical factors to develop more accurate risk stratification models for post-stroke cognitive impairment. While our study lays the groundwork for the potential use of HMGB1 as a prognostic biomarker in stroke patients, further large-scale, multi-center studies and replication efforts are needed to confirm and extend our findings. Such research will be essential for translating these results into clinical applications and improving the management of patients at risk for post-stroke cognitive impairment.

In addition to these limitations related to the study design, it is important to acknowledge the potential biases introduced by the subjective nature of the NIHSS and MoCA scales used in our study. We acknowledge that the NIHSS and MoCA are subjective scales, which may introduce potential biases in the assessment of neurological deficits and cognitive function. To minimize these biases, we employed several strategies, including standardized training for evaluators, blinding of neuropsychologists to clinical information, and regular quality control measures to ensure consistency in administration and scoring. Despite these efforts, we recognize that some degree of subjectivity is inherent in these assessments, which may limit the generalizability of our findings. Future studies may benefit from the use of additional objective measures, such as neuroimaging or performance-based cognitive tests,

to complement the subjective scales and provide a more comprehensive assessment of neurological and cognitive function.

The results of this study suggest that elevated early serum HMGB1 levels are associated with an increased risk of cognitive decline after stroke, which has important implications for future research. Firstly, these findings support further investigation into HMGB1 as a prognostic biomarker for PSCI in larger validation cohorts and long-term longitudinal studies. If confirmed, measurement of serum HMGB1 could be incorporated into clinical evaluation of stroke patients to identify those requiring closer monitoring and preventative treatment for cognitive dysfunction. Secondly, more research is needed to elucidate the mechanisms by which heightened HMGB1 contributes to secondary neurodegeneration and cognitive impairment after ischemic stroke. This could uncover novel therapeutic targets for pharmacological inhibition of HMGB1 signaling to mitigate cognitive decline. Thirdly, since HMGB1 levels can be modulated by treatments like thrombolysis, studies could examine whether therapies that reduce HMGB1 release also lower risk of later cognitive problems. Finally, HMGB1 may interact with other markers of neuroinflammation and biomarkers of nerve injury-a combined biomarker set that could provide greater predictive and prognostic utility for PSCI and warrants further investigation. In conclusion, these findings open promising new avenues of research to confirm HMGB1 as a clinically viable biomarker and to elucidate its role in cognitive prognosis after ischemic stroke.

## CONCLUSION

In this prospective cohort investigation, heightened concentrations of HMGB1 in the serum during the acute phase following ischemic stroke emerged as an independent predictor of cognitive dysfunction assessed at the 3-month follow-up time point. Patients who developed cognitive impairment had significantly higher initial HMGB1 levels. HMGB1 showed a dose-dependent relationship with risk of cognitive impairment. These results suggest that serum HMGB1 could potentially serve as a valuable blood-based biomarker for prognostic stratification, allowing clinicians to identify individuals with ischemic stroke who are at higher risk of experiencing cognitive deterioration in the short-term following the event. Additional studies with larger patient populations are needed to validate the utility of HMGB1 as a predictive biomarker for cognitive decline following acute ischemic stroke.

### Funding

This work was supported by the Shandong Medical and Health Technology Development Fund (No. 202103070325), the Shandong Province Traditional Chinese Medicine Science and Technology Project (No. M-2022216) and the Nursery Project of the Affiliated Tai'an City Central Hospital of Qingdao University (No. 2022MPM06). The funders had no role in study design, data collection and analysis, decision to publish, or preparation of the manuscript.

## Grant Disclosures

The following grant information was disclosed by the authors:

The Shandong Medical and Health Technology Development Fund: No. 202103070325.

The Shandong Province Traditional Chinese Medicine Science and Technology Project: No. M-2022216.

The Nursery Project of the Affiliated Tai'an City Central Hospital of Qingdao University: No. 2022MPM06.

## Competing Interests

The authors declare there are no competing interests.

## Author Contributions

- Zhenbao Liu performed the experiments, analyzed the data, prepared figures and/or tables, authored or reviewed drafts of the article, and approved the final draft.
- Weixia Yang performed the experiments, analyzed the data, prepared figures and/or tables, authored or reviewed drafts of the article, and approved the final draft.
- Jianxin Chen conceived and designed the experiments, analyzed the data, prepared figures and/or tables, authored or reviewed drafts of the article, and approved the final draft.
- Qian Wang conceived and designed the experiments, analyzed the data, prepared figures and/or tables, authored or reviewed drafts of the article, and approved the final draft.

## Human Ethics

The following information was supplied relating to ethical approvals (*i.e.,* approving body and any reference numbers):

The Ethics Committee of Jinan First People's Hospital.

## Data Availability

The raw measurements are available in the Supplementary File.

## Supplemental Information

Supplemental information for this article can be found online at http://dx.doi.org/10.7717/peerj.17309#supplemental-information.

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
