# Peer review of "Circulating HMGB1 in acute ischemic stroke and its association with post-stroke cognitive impairment"

_PeerJ, doi:10.7717/peerj.17309_

## Round 0.1 · original submission · Major Revisions

The authors should carefully answer these concerns.

**Language Note:** The review process has identified that the English language must be improved. PeerJ can provide language editing services - please contact us at [email protected] for pricing (be sure to provide your manuscript number and title). Alternatively, you should make your own arrangements to improve the language quality and provide details in your response letter. – PeerJ Staff

Reviewer 1 ·

Basic reporting

The strength of this study lies in its prospective design and focus on early measurement of serum HMGB1 within 24 hours of stroke onset. The novelty of this approach is emphasized and how it contributes to the current understanding of HMGB1 as a potential biomarker for PSCI.

Experimental design

This study establishes an association between early elevated HMGB1 levels and PSCI. However, the manuscript may benefit from a more nuanced discussion of the challenges of inferring causality from observational studies.

Validity of the findings

Discusses the importance of validation in larger cohorts or replication studies given the single-center nature of this study and the relatively small sample size. Emphasize the need for further studies to confirm the findings.

Additional comments

Although this study delved into the relationship between HMGB1 and PSCI, its potential interactions with broader neuroinflammatory processes should have been detailed.
In conclusion, this study provides valuable insights into the potential predictive role of HMGB1 in PSCI. Strengthening the discussion of association and causation, and emphasizing the need for validation and further research will enhance the impact of the manuscript.

Reviewer 2 ·

Basic reporting

This prospective cohort study investigated the potential of serum high mobility group box 1 (HMGB1) levels within 24 hours after acute ischemic stroke as a predictor of 3-month poststroke cognitive impairment (PSCI). This study was well designed with clear objectives and methods. The manuscript is comprehensive and provides detailed information on patient selection, data collection, and statistical analysis. The findings suggest a significant association between elevated HMGB1 levels and increased risk of PSCI. This study provides valuable insights for early identification of patients at risk for cognitive decline after stroke.

Experimental design

(1) Rigorous research methodology:The study design, patient selection, and data analysis were well documented, increasing the credibility of the results.
(2) Novelty:The study fills a gap in research by focusing on early HMGB1 levels and their association with PSCI, adding valuable information to the field. The findings suggest that serum HMGB1 could be a useful prognostic biomarker for identifying stroke patients at risk for short-term cognitive decline.

Validity of the findings

(1) Larger sample size: While the study provided valuable insights, a larger sample size would enhance the generalizability of the findings. Consider discussing the potential impact of sample size on study power and generalizability.
(2) Validation Cohort:The lack of a validation cohort limits the robustness of the findings. Consider addressing this limitation in the discussion and suggest future validation cohort studies.
(3) Causality: Although the study identified an association between HMGB1 and PSCI, it is important to emphasize that the observational nature of the study does not allow for causality to be established. This should be discussed in the limitations section.

Additional comments

The manuscript needs to be revised by a fluent English speaker to make it more readable.

Reviewer 3 ·

Basic reporting

PSCI is the most common complication after stroke, causing many inconveniences and affecting the life treatment of stroke patients.The mechanism of PSCI is also not well understood, and this manuscript suggests that HMGB1 can be used as a predictive marker with some potential applications.
1. The authors mention that "blood specimens were collected within 24 hours of fasting", how long was the fasting period?
2. The NHISS and MoCA are highly subjective as scales, how do you balance bias?
3. The authors mention that "NHISS" is authorized. As far as I know, NHISS does not need to be licensed, are you referring to MoCA? Please check further.

Experimental design

.

Validity of the findings

.

---

## Round 0.2 · accepted · Accept

All the reviewers have accepted the manuscript.

Reviewer 1 ·

Basic reporting

satisfied

Experimental design

satisfied

Validity of the findings

satisfied

Additional comments

satisfied

Reviewer 2 ·

Basic reporting

This prospective cohort study investigated the potential of serum high mobility group box 1 (HMGB1) levels within 24 hours after acute ischemic stroke as a predictor of 3-month poststroke cognitive impairment (PSCI). This study was well designed with clear objectives and methods. The manuscript is comprehensive and provides detailed information on patient selection, data collection, and statistical analysis. The findings suggest a significant association between elevated HMGB1 levels and increased risk of PSCI. This study provides valuable insights for early identification of patients at risk for cognitive decline after stroke.

Experimental design

The study design, patient selection, and data analysis were well documented.

Validity of the findings

Conclusions are well stated. The findings suggest that serum HMGB1 could be a useful prognostic biomarker for identifying stroke patients at risk for short-term cognitive decline.

Additional comments

none

Reviewer 3 ·

Basic reporting

The authors have addressed my concerns well.

Experimental design

The authors have addressed my concerns well.

Validity of the findings

The authors have addressed my concerns well.

Additional comments

The authors have addressed my concerns well.